# A Computational Approach to Sequential Decision Optimization in Energy Storage and Trading

Paolo Falbo [1] , Juri Hinz [2,*] , Piyachat Leelasilapasart [2] and Cristian Pelizzari [1]

1 Department of Economics and Management, University of Brescia, 25122 Brescia, Italy; paolo.falbo@unibs.it (P.F.); cristian.pelizzari@unibs.it (C.P.)
2 School of Mathematical and Physical Sciences, University of Technology Sydney, P.O. Box 123, Ultimo, NSW 2007, Australia; piyachat.leelasilapasart-1@student.uts.edu.au
* Correspondence: juri.hinz@uts.edu.au

**Abstract:** Due to recent technical progress, battery energy storages are becoming a viable option in the power sector. Their optimal operational management focuses on load shift and shaving of price spikes. However, this requires optimally responding to electricity demand, intermittent generation, and volatile electricity prices. More importantly, such optimization must take into account the so-called deep discharge costs, which have a significant impact on battery lifespan. We present a solution to a class of stochastic optimal control problems associated with these applications. Our numerical techniques are based on efficient algorithms which deliver a guaranteed accuracy.

**Keywords:** approximate dynamic programming; energy trading; optimal control; power sector



## 1. Introduction

In many countries, Battery Energy Storage Systems (BESS) are becoming popular due to their advantages in managing power dispatch, interconnection, and demand. Their growing acceptance is due to their ability to smooth out the intermittent and unreliable nature of Renewable Energy Sources (RES). Although RES penetration has shown an increasing trend, their unreliable energy supply makes it difficult to incorporate RES into a modern electricity grid. However, in some niche applications, a variety of BESS is already installed, where they provide operational efficiency and reduce costs by exploiting synergies between storage and renewables (Barton and Infield 2004; Black and Strbac 2007; Kim and Powell 2011; Teleke et al. 2010). Nonetheless, the implementation of BESS is still not straightforward from economical and technical perspectives because of BESS costs and their sensitivity to deep discharge. In this domain, there is a large body of literature with diverse contributions to economic and technological aspects of BESS management. Furthermore, we emphasize Yang et al. (2014) and Kempener and Borden (2015), investigating the role of batteries with respect to RES, and Lu et al. (2014), on the optimal use of BESS for the so-called peak load shaving.

Let us describe an abstract, but typical framework for BESS application. The traditional electricity market players satisfy consumers' energy demand by purchasing electricity in advance, usually taking positions in the *long-term market*. Long-term market may stand for any energy delivery agreements purchased prior to the delivery period (a year, a semester, a month), depending on the situation. However, this market is typically represented by the so-called day-ahead market for hourly delivery on the next day. On the contrary, the imbalances during the delivery period must be compensated, as they occur, at a *short-term market*. Such short-term energy balancing can either be achieved through complex over-the-counter trading or, more realistically, by participating in real-time energy auction, or by transferring supply from or to electricity grid at the so-called real time grid prices.

Figure 1 provides a simplified illustration of this optimal problem for a producer endowed with renewable generation capacity. However, in the presence of storage and

renewable generation facilities, the problem changes. Within this framework, the agent is now required to simultaneously take long-term positions and setting energy storage levels, as shown in Figure 2. The decision optimization problem becomes significantly more complex due to the uncertainty stemming from the future battery levels, electricity prices, and output of renewable energy.

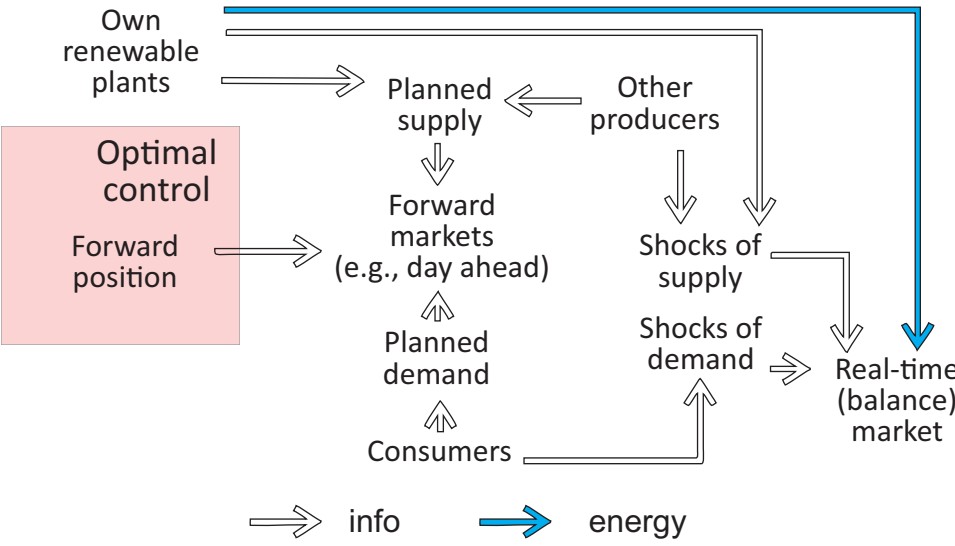

**Figure 1.** Traditional energy dispatch. White arrows represent flows of information, whereas filled arrows stand for flows of energy. The matching of planned demand with energy supply on the forward market is not depicted.

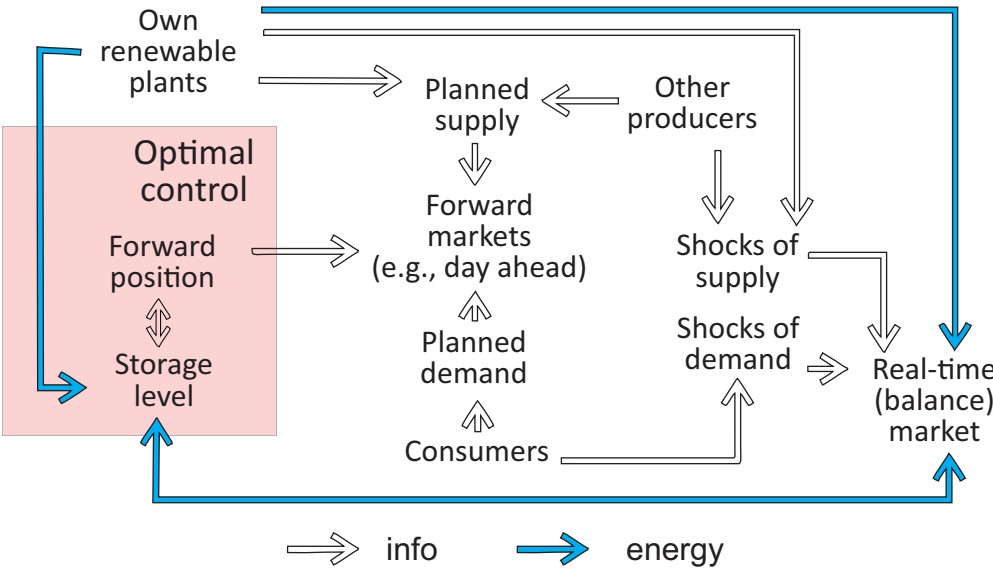

**Figure 2.** Energy dispatch in the presence of renewable energy and battery storage. White arrows represent the flows of information, whereas filled arrows stand for flows of energy. The matching of planned demand with energy supply on the forward market is not depicted.

Typically, renewable energy sources, such as wind and solar, are notoriously intermittent and unreliable. The potential of energy storages to address the highly intermittent nature of renewable energy generation Breton and Moe (2009); Dincer (2011) and energy demand has been discussed in the literature (see Beaudin et al. (2010); Diaz-Gonzalez et al. (2012); Evans et al. (2012); Kempener and Borden (2015); Yang et al. (2014)). Their incorporation into a modern energy grid will encourage more environmentally friendly policies that will

also have a significant impact on investor atittudes towards firms Chan and Walter (2014); Ramiah et al. (2013); Renneboog et al. (2008). The authors of Lu et al. (2014) studied the possible usage of battery storage systems to defer costly modifications to the energy grid by addressing peak loads in the power grid. A recent review of available energy storage technologies has been given by Luo et al. (2015) and Weitzel and Glock (2018).

While there exist numerous types of energy storage systems, Beaudin et al. (2010) found that no single storage system consistently outperforms all of the others for all types of renewable energy sources and applications. Hence, for the sake of simplicity, this paper will assume that the energy retailer pictured in Figure 2 uses a battery device for storing energy. However, our methods and results can easily be extended to other types of storage technologies, or even to the use of multiple types of storage devices. From a real options analysis viewpoint (applied in this context), the incorporation of energy storages into the energy grid also poses interesting investment questions. The work Bakke et al. (2016); Bradbury et al. (2014); Locatelli et al. (2016) examines the profitability of investing (in energy storage), while Schachter and Mancarella (2016) questions the suitability of the real options approach, stating that the risk neutrality assumption may not be appropriate.

The integration of electric energy storage systems yields challenging problems of optimal stochastic control type (see Dokuchaev and Zolotarevich (2020); Kim and Powell (2011); Oudalov et al. (2007); Teleke et al. (2010), among others). While the traditional power generation considers a sequence of independent decisions, the opportunity of storing electricity intertwines actions made at different times: It is obvious that storage facilities can be charged during the base load low-price hours and discharging during peak load high-price hours. However, how exactly to do this requires mathematical techniques, i.e., dynamic programming (AzRISE 2010; Löhndorf and Minner 2010), or the calculus of variations (Flatley et al. 2016). Apart from rare cases, where a dynamic programming can be addressed analytically (Bäuerle and Rieder 2011; Pham 2009), these problems are usually based on numerical techniques and they are addressed by approximate dynamic programming methods (Powell 2007). Although a huge variety of computational tools has been developed in this area, real-world problems are too complex for existing solution techniques, since the number of decision factors is high. The present contribution extends the state of the art in several aspects.

The first aspect is about modeling a complex financial and economic environment, which electricity retailers are routinely confronted with. We consider energy trading on two time scales: a long-term (realized as day-ahead trading, futures, or forward contracts) and a short-term (designed to adjust unexpected changes in the electricity demand in front of delivery, which can be realized by the so-called intra-day trading, or by diverse energy balancing procedures). Such a structure is typical for all energy markets, with differences in price dynamics, liquidity, and spreads. The distinction between the short-term and the long-term energy trading is of primary importance to correctly determine the economic value of a BESS. The present work extends that of Hinz and Yee (2018a), who address BESS management within an over-simplistic setting, merely considering a battery installation as a passive buffer, which settles energy imbalance against electricity grid. In difference to this, our work makes a realistic assumption that the battery levels and energy trading are simultaneously controlled, placing the costs of deep discharge at the core of investigation.

The second aspect is methodological: we attempt solving a class of battery management problems, rather than a specific case. More precisely, here we present a computational methodology, whose routines implement stochastic switching algorithms (written in the scientific language Julia.[1]) Our approach realizes a highly customizable solution. Namely, the entire source code of our computations (available on Github) comprises several blocks that serve as place holders and can be tailored to specific situations. For instance, the state space model for electricity prices (including seasonal and mean-reverting components) considered in this work is not attempted to describe any typical electricity price pattern, being a proxy that can be replaced, modified, and adjusted. Such flexibility is ensured by assumptions on linear state dynamics that encompass any ARMA model combined

with appropriate seasonal and trend components. The same considerations apply to the modeling of deep discharge costs. We suggest a simple proxy function penalizing deep discharge in a fairly general way allowing for the user to tailor such penalization to a particular battery technology.

The third aspect regards a novel computation technique. In comparison to existing schemes (for instance, Löhndorf and Minner (2010)), we optimize energy storage using a combination of primal and dual schemes. More specifically, we apply a sub-gradient method introduced by Hinz (2014); Hinz and Yap (2015); Hinz and Yee (2018a) to obtain, in a first step, an approximate solution of our stochastic control problem whose numerical quality is examined, in the second step, using a dual method (Hinz et al. 2020; Hinz and Yap 2015). That is, we provide simultaneous optimization of battery control and long-term trading in presence of the uncertainty that results from RES generation, demand, and electricity price with *guaranteed precision*, which contrasts this work from all existing contributions, to the best of authors knowledge.

The paper is organized, as follows: Section 2 details the model settings. Section 3 briefly reviews the adopted solution technique. Section 4 applies the solution technique to the present decision problem. Section 5 includes an illustrative case study, while Section 6 concludes.

## 2. Model Settings

In the following, we present an abstract, but generic, model for an electricity market, where an energy retailer has the obligation to meet the demand of its consumers using a combination of energy from renewable generation, contractual position from long-term market (for instance, day-ahead trading), and short-term market (real-time trading, balancing market) as well as battery storage.

Because, in reality, decisions are made and revised periodically, we propose a dynamic optimization with discrete-time decision-making. The present framework encompasses all important features of real-world energy trading in order to illustrate our methodology in the most general setting. Following this approach, our algorithmic solution can be adjusted to a specific market architecture.

We assume a given finite-time horizon $\{0, \ldots, T\} \subset \mathbb{N}$ and agree that the unit time corresponds to the energy delivery interval, which can measure hours, days, or weeks, depending on the particular application. At any time $t = 0, \ldots, T - 1$, an energy retailer has the obligation to satisfy, within $[t, t + 1]$, the unknown electricity demand of its customers, while the retailer's renewable energy sources produce a random electricity amount. The retailer must trade electricity optimally in advance (at time $t$) and decide how to control the battery in order to manage the resulting energy imbalance.

The revenue optimization of such retailer is a sequential decision problem under uncertainty. That is, at any time $t = 0, \ldots, T - 1$, an action must be chosen (that will encompass both energy trading and BESS control). Such action also influences the transition to the subsequent state (next battery levels), changing all future revenues, costs, and decisions. The minimization of control costs in such setting is naturally addressed in terms of the so-called Markov Decision Theory. In the subsequent paragraphs, we formulate our optimization problem within this framework, which we customize accordingly to be ably applying our approximate solution methodology.

Let $D_{t+1}$ denote the residual electricity demand after that all renewable generation has been sold to customers. Here, $D_{t+1} > 0$ stands for shortfall and $D_{t+1} < 0$ for excess in the delivery period $[t, t + 1]$ beginning at time $t$. We assume that the random variable $D_{t+1}$ is observed at $t + 1$ when the delivery period $[t, t + 1]$ ends. Assume that, at each time $t = 0, \ldots, T - 1$, the producer can take a position $F_t$ in a long-term market to ensure energy supply from the market to the retailer (if $F_t > 0$) or outflow from the retailer to the market (if $F_t < 0$) within $[t, t + 1]$. Further consider the variable $B_t$, which stands for the decision to use the battery energy $|B_t|$, where $B_t > 0$ and $B_t < 0$ represent the charging

and discharging actions, respectively. Here, we agree that the BESS control actions must be decided at time $t$, before the start of the delivery period.

With these assumptions, the energy to be balanced within $[t, t+1]$ through the short-term market that is given by

$$F_t - D_{t+1} - B_t. \tag{1}$$

Let us introduce the random variables

$$\psi_t, \qquad t = 0, \ldots, T, \quad \underline{\Psi}_{t+1}, \quad \overline{\Psi}_{t+1}, \qquad t = 0, \ldots, T-1, \tag{2}$$

that stand for prices of electricity delivered within the time interval $[t, t+1]$. Thereby, we assume that $\psi_t$ is the long-term market price (that is observed and paid at $t$ for energy delivered as within $[t, t+1]$), whereas $\underline{\Psi}_{t+1}$ and $\overline{\Psi}_{t+1}$ represent the short-term market prices. Here, $\underline{\Psi}_{t+1}$ applies for procurement and $\overline{\Psi}_{t+1}$ for the purchase of energy delivered as constant flow within $[t, t+1]$. Note that both short-term market prices $\underline{\Psi}_{t+1}$, $\overline{\Psi}_{t+1}$ are not observable at the time $t$ and become known at $t+1$ at the end of period $[t, t+1]$.

**Remark 1.** *The relations between prices of electricity, delivered at different time scales and traded at different market places, has been lively discussed since beginning of energy market deregulation. Naturally, the connection between long-term (day-ahead) and the short-term (balancing auction, real-time) prices heavily depends on risk aversion, variable production costs, production capacities, and delivery commitments of all market participants. For an overview on equilibrium analysis, we refer an interested reader to Hinz (2003) and the literature cited therein.*

Finally let us denote, by $C_t$, the operational costs that are associated with battery management, which must be modeled by a random variable, depending on the recent battery level and on energy delivered/absorbed within $[t, t+1]$. This quantity will reflect the impact of deep discharge on battery life, expressed in monetary units. At the moment, we postpone the specification of these costs and finalize a one-period revenue from BESS management, as

$$R_t = -F_t \psi_t - (F_t - D_{t+1} - B_t)^- \overline{\Psi}_{t+1} + (F_t - D_{t+1} - B_t)^+ \underline{\Psi}_{t+1} - C_t$$

for $t = 0, \ldots, T-1$. In order to maximize the expectation of the total revenue

$$R = \sum_{t=0}^{T-1} R_t, \tag{3}$$

a stochastic control problem must be solved. This problem is dynamical in the sense that, at any time $t$, the decision to charge/discharge battery changes the situation at the next decisions time $t+1$, which has a profound effect on the next-period planning. Typically, such a sequential decision problem admits no closed-form solution, and it can be computationally challenging.

It is well-known that the demand fluctuations follow a complex seasonal pattern and they are difficult to model, particularly by Markovian processes required for the state dynamics. At this point, we suggest a significant simplification: it turns out that, under generic assumptions, the demand modeling can be split off from the strategy optimization. We show that merely one-step prediction $d_t$ (conditional expectation on the most recent information) of the energy demand $D_{t+1}$ is relevant. That is, the problem that is addressed in this work separates into two distinct steps:

(i)   Establishing a time-series model for the dynamics of the energy demand that serves at any time $t$ the conditional expectation $d_t$ of the demand $D_{t+1}$ occurring within $[t, t+1]$.

(ii) Solving the stochastic dynamic control problem, where the demand prediction $d_t$ is not a decision variable at time $t$, because the optimal long-term position is calculated as a deviation from the demand prediction.

(iii) Running an optimal policy. For this, the prediction of the demand must be available.

Note that the second step is disentangled in the sense that, for (ii), the dynamics of the demand prediction is irrelevant. On this account, we only consider (ii) in the reminder of this work. However, notice that, for strategy implementation in (iii), the demand prediction must be available at any decision time. However an advantage is that the user can alter or replace the entire demand prediction model without re-calculating the sequential decision strategy.

Let us establish such an approach and make some assumptions to obtain a model that can be solved by our numerical methodology. We suppose that there is finite set $A$ of all possible actions. At any time, the controller chooses an action via an (optimal) decision rule that we determine later. The actions can be defined in an abstract way, or they can be identified by integers (or indexed by integer vectors), and their meaning for control is merely established by some functions that are defined on actions $A$. In the literature, such functions are known as look-up tables. In our case, to model diverse choices of the trade volume in the long-term market, a function $f : A \to \mathbb{R}$ is used with the following interpretation: given the prediction $d_t$ of the demand $D_t$ occurring within $[t, t+1]$, for the action $a_t \in A$ chosen at time $t$ by the controller, the energy volume that is traded in advance (long-term market) is

$$F_t = f(a_t) + d_t. \tag{4}$$

The quantity $|f(a_t)|$ is the energy (bought if $f(a_t) > 0$, sold if $f(a_t) < 0$) on the top of the predicted demand $d_t$ and it will be referred to as a *safety margin*. The function $f$ must be chosen in advance by the decision maker and it typically consists of fixing both the granularity and range for safety margins.

Similarly, the battery management variable $B_t$ is also determined by the action $a_t$ chosen at time $t$ (immediately before the start of delivery period $[t, t+1]$). Here, we again use an appropriate function on $A$, but this time the modeling is more complex, as the energy absorbed/delivered by the battery must take the current battery level and the physical constraints into account. To detail this, we suggest discretizing the battery levels by a finite set $P$. Having chosen action $a_t \in A$ at time $t = 0, \ldots, T-1$, we suppose that the current battery level $p_t \in P$ transforms to the next level $p_{t+1} = \ell(p_t, a_t) \in P$ in terms of a pre-specified level change function $\ell : P \times A \to P$ representing the technical restrictions of the battery (total capacity, electrical power). For instance, $a_t \mapsto l(p_t, a_t)$ can have values above and below $p_t$ within a range representing one-period charge/discharge power restrictions, if $p_t$ is in one of the intermediate battery levels. However, if $p_t$ is the highest (the lowest) level, then $a_t \mapsto l(p_t, a_t)$ only take values below (above) $p_t$. Specifying the function $l$ requires some details of battery technology used, in particular for determining the maximal charge/discharge intensity along with the highest and lowest (admissible) battery levels.

Notice that, with this convention, the energy amount transformed from/to the storage is given by $B_t = b(p_t, a_t)$, where

$$b(p_t, a_t) = \begin{cases} \ell(p_t, a_t) - p_t, & \text{if } \ell(p_t, a_t) - p_t > 0 \\ \kappa \cdot (\ell(p_t, a_t) - p_t), & \text{if } \ell(p_t, a_t) - p_t \leq 0 \end{cases}, \tag{5}$$

with the constant $\kappa \in [0, 1]$ standing for battery efficiency. With these assumptions, we express the energy imbalance (1) in terms of action $a_t \in A$ and battery level $p_t \in P$ while using prediction error

$$\varepsilon_{t+1} = D_{t+1} - d_t, \qquad t = 0, \ldots, T-1$$

as

$$
\begin{aligned}
F_t - D_{t+1} - B_t &= d_t + f(a_t) - (d_t + \varepsilon_{t+1}) - b(p_t, a_t) \\
&= f(a_t) - b(p_t, a_t) - \varepsilon_{t+1}.
\end{aligned} \tag{6}
$$

That is, an action $a_t \in A$ not only triggers transition in battery level from $p_t$ to $p_{t+1} = \ell(p_t, a_t)$, but it also determines the the energy amount that needs to be balanced against at the short-term market. Having defined the excess and shortage of the imbalance (6) by

$$
\begin{aligned}
\overline{E}_{t+1}(a_t, p_t) &= (f(a_t) - b(p_t, a_t) - \varepsilon_{t+1})^- \\
\underline{E}_{t+1}(a_t, p_t) &= (f(a_t) - b(p_t, a_t) - \varepsilon_{t+1})^+
\end{aligned}
$$

the profit/loss from balancing at time $t = 0, \dots, T - 1$ is modeled by

$$
- \overline{E}_{t+1}(a_t, p_t) \overline{\Psi}_{t+1} + \underline{E}_{t+1}(a_t, p_t) \underline{\Psi}_{t+1}. \tag{7}
$$

Now, using (4), the financial position for action $a_t \in A$ is

$$
F_t \psi_t = d_t \psi_t + f(a_t) \psi_t \tag{8}
$$

(where $F_t \psi_t > 0$ is interpreted as the cost to guarantee energy from the long-term market.)

Finally, let us model the storage costs by

$$
C_t = c(p_t, a_t), \tag{9}
$$

reflecting the dependence on action $a_t \in A$ and battery level $p_t \in P$. The details of the costs must be described by an appropriate function

$$
c : P \times A \to \mathbb{R}
$$

specified in accordance to battery technology.

With the assumptions (7), (8), and (9), the profit/loss that is associated with action $a_t \in A$ depends on prices $\psi_t, \overline{\Pi}_{t+1}, \underline{\Pi}_{t+1}$, the demand prediction $d_t$, and battery level $p_t \in P$ as

$$
R_t = -d_t \psi_t - f(a_t) \psi_t - \overline{E}_{t+1}(a_t, p_t) \overline{\Psi}_{t+1} + \underline{E}_{t+1}(a_t, p_t) \underline{\Psi}_{t+1} - c(p_t, a_t).
$$

Observe that the term $d_t \psi_t$ depends neither on the action $a_t$ nor on the battery level $p_t$. Because this quantity can not be changed by the decision optimization, the action-dependent part of the reward is modeled in terms of

$$
\begin{aligned}
\tilde{R}_t &= R_t + d_t \psi_t \\
&= -f(a_t) \psi_t - \overline{E}_{t+1}(a_t, p_t) \overline{\Psi}_{t+1} + \underline{E}_{t+1}(a_t, p_t) \underline{\Psi}_{t+1} - c(p_t, a_t).
\end{aligned} \tag{10}
$$

In what follows, we show how to obtain a strategy that simultaneously takes positions on long-term market and controls a battery level to maximize the expectation of the demand-adjusted total revenue

$$
\tilde{R} = \sum_{t=0}^{T-1} \tilde{R}_t. \tag{11}
$$

**Remark 2.** *Note that (11) differs from (3) by a random variable $\sum_{t=0}^{T-1} d_t \psi_t$, not depending on the control policy. On this account, maximizing the expectation of $\tilde{R}$ constitutes a solution to our problem. As mentioned above, such an approach avoids tedious modelling of energy demand dynamics. However, notice that the results cannot be used directly: for strategy implementation, the*

*one-step demand prediction must be available at any time, thus a demand model is required to run the policy.*

## 3. Research Methodology and Solution Techniques

This paper utilizes a novel numerical technique for sequential decision optimization, which requires a number of specific assumptions. To make this technique applicable, we model the battery storage optimization. In what follows, we present, in detail, our methodology and elaborate on the assumptions required. However, to give the reader an orientation, let us highlight some of the most important aspects beforehand. The technique represents a combination of a dual and a primal approach. Thereby the primal methodology delivers an approximate numerical solution whose quality is examined using duality methods. The primal solution is based on a specific function approximation method requiring convexity. Linear state dynamics is assumed in order to retain convexity through the backward induction. The dual part is based on solution diagnostics that can be viewed as Monte Carlo-based backtesting with variance reduction. This technique has been used extensively for optimal stopping problems and it has been extended to our context. Because of the combination of primal and a dual methods, the methodology delivers high performance with guaranteed accuracy.

Sequential decision making is usually encompassed by discrete-time stochastic control and it is addressed by *Markov Decision Processes/Dynamic Programming*. This theory provides a variety of methods. However, approaching analytical solutions may be cumbersome (Bäuerle and Rieder 2011; Pham 2009; Powell 2007) and numerical approximations may often be far more practical. This work will utilize an implementation of fast and accurate algorithms (see Hinz and Yee 2018b) to address specific control problems, assuming a finite-time horizon, a finite set of actions, convex reward functions, and a state process following a linear dynamics. Although these assumptions are restrictive, they encompass a large class of practically important control problems and yield approximate solutions with excellent precision and numerical performance. Let us briefly describe this approach.

Suppose that the state space $P \times \mathbb{R}^d$ is a Cartesian product of a finite set $P$ and $\mathbb{R}^d$. Furthermore, assume that a finite set $A$ represents all possible actions. Given a finite-time horizon $\{0, \ldots, T\} \subset \mathbb{N}$, consider a *fully observable* controlled Markovian process $(X_t)_{t=0}^{T} := (P_t, Z_t)_{t=0}^{T}$ that consists of two parts.

**Stochastic switching:** Referrers to the evolution of the discrete component $(P_t)_{t=0}^{T}$, which is described by a finite-state controlled Markov chain, taking values in a finite set $P$. This means that, at any time $t = 0, \ldots, T-1$, the controller chooses an action $a$ from $A$ in order to trigger the one-step transition from the mode $p \in P$ to the mode $p' \in P$ with probability $\alpha_{p,p'}(a)$, where $(\alpha_{p,p'}(a))_{p,p' \in P}$ are pre-specified transition probability matrices for all $a \in A$.

**Linear dynamics:** referrers to the evolution of the continuous component $(Z_t)_{t=0}^{T}$, which is assumed to follow an uncontrolled evolution of such a component in the Euclidean space $\mathbb{R}^d$. The evolution is modeled by the recursion

$$Z_{t+1} = W_{t+1} Z_t, \qquad t = 0, \ldots, T-1, \tag{12}$$

where $(W_t)_{t=1}^{T}$ are independent *disturbance matrices*.

**State evolution:** that is, the transition kernels $\mathcal{K}_t^a$ governing the evolution of our controlled Markovian process $(P_t, Z_t)_{t=0}^{T}$ from time $t$ to time $t+1$ are given, for each $a \in A$, by

$$\mathcal{K}_t^a v(p,z) = \sum_{p' \in P} \alpha_{p,p'}(a) \mathbb{E}(v(p', W_{t+1}z)), \quad p \in P, z \in \mathbb{R}^d, t = 0, \ldots, T-1,$$

that acts on each function $v : P \times \mathbb{R}^d \to \mathbb{R}$, where the above expectations are well-defined.

**Costs of control:** if the system is in the state $(p, z)$, the *rewards* of applying action $a \in A$ at

time $t = 0, \dots, T - 1$ are given by $r_t(p, z, a)$. Having arrived at time $t = T$ in the state $(p, z)$, a final *scrap value* $r_T(p, z)$ is collected. Thereby, the reward functions $r_t : P \times \mathbb{R}^d \times A \to \mathbb{R}$, as well as the scrap function $r_T : P \times \mathbb{R}^d \to \mathbb{R}$, are exogenously given for $t = 0, \dots, T - 1$. At each time $t = 0, \dots, T - 1$, the *decision rule* $\pi_t$ is given by a mapping $\pi_t : P \times \mathbb{R}^d \to A$, prescribing at $t$ an action $\pi_t(p, z) \in A$ in a state $(p, z) \in P \times \mathbb{R}^d$. Note that, at each time, the decision rule refers to the recent state of the system, representing a so-called *feedback control*. A sequence $\pi = (\pi_t)_{t=0}^{T-1}$ of decision rules is called a *policy*. For each policy, $\pi = (\pi_t)_{t=0}^{T-1}$, the policy value $v_0^\pi(p_0, z_0)$ is defined as the total expected reward

$$v_0^\pi(p_0, z_0) = \mathbb{E}^{(p_0, z_0), \pi}\left[\sum_{t=0}^{T-1} r_t(P_t, Z_t, \pi_t(P_t, Z_t)) + r_T(P_T, Z_T)\right].$$

In this formula, $\mathbb{E}^{(p_0, z_0), \pi}$ stands for the expectation with respect to the probability distribution of $(P_t, Z_t)_{t=0}^{T}$ that is defined by Markov transitions from $(P_t, Z_t)$ to $(P_{t+1}, Z_{t+1})$ that are induced by the kernels $\mathcal{K}_t^{\pi_t(P_t, Z_t)}$ for $t = 0, \dots, T - 1$, started at the initial point $(P_0, Z_0) = (p_0, z_0)$.

**Optimization goal:** a policy $\pi^* = (\pi_t^*)_{t=0}^{T-1}$ is called optimal if it maximizes the total expected reward over all policies $\pi \mapsto v_0^\pi(p, z)$. To obtain such a policy, one introduces, for $t = 0, \dots, T - 1$, the so-called *Bellman operator*

$$\mathcal{T}_t v(p, z) = \max_{a \in A}\left[r_t(p, z, a) + \sum_{p' \in P} \alpha_{p,p'}(a)\mathbb{E}[v(p', W_{t+1}z)]\right], \quad \text{for } (p, z) \in P \times \mathbb{R}^d, \quad (13)$$

acting on all functions $v$ where the stochastic kernel is well-defined. Consider the *Bellman recursion*, which is also referred to as backward induction:

$$v_T = r_T, \quad v_t = \mathcal{T}_t v_{t+1} \qquad \text{for } t = T - 1, \dots, 0. \qquad (14)$$

Assuming that the reward functions are convex and globally Lipschitz (in the second variable) and the disturbance matrices $(W_t)_{t=1}^{T}$ are integrable, there exists a solution $(v_t^*)_{t=0}^{T-1}$ to the Bellman recursion. Such functions $(v_t^*)_{t=0}^{T-1}$ are called *value functions*, they determine an optimal policy $\pi^* = (\pi_t^*)_{t=0}^{T-1}$ via

$$\pi_t^*(p, z) = \arg\max_{a \in A}\left[r_t(p, z, a) + \sum_{p' \in P} \alpha_{p,p'}(a)\mathbb{E}[v_{t+1}^*(p', W_{t+1}z)]\right], \qquad (15)$$

for $t = T - 1, \dots, 0$ and $v_T^* = r_T$.

**Remark 3.** *In applications, sequential decision problems frequently appear in a slightly different formulation than given above. Usually, the costs of control depend on both the recent and the next state. That is, instead of a previously introduced reward $r_t(P_t, Z_t, a)$ for taking action $a$ in the situation $(P_t, Z_t)$, a modeling may naturally suggest $\tilde{r}_t(P_t, Z_t, P_{t+1}, Z_{t+1}, a)$ where the action $a$ is taken at time $t$ in the situation $(P_t, Z_t)$ but reward is observed and returned at $t + 1$ with a random outcome depending on the next-time situation $(P_{t+1}, Z_{t+1})$. Fortunately, this context is seamlessly covered by the formal setting introduced above. It turns out that, since the expectation of reward is being maximized, a pre-conditioning $\tilde{r}_t(P_t, Z_t, P_{t+1}, Z_{t+1}, a)$ on the information that is available at time $t$ can be applied. That is, having determined the control rewards as being next-state dependent*

$$\tilde{r}_t : P \times \mathbb{R}^d \times P \times \mathbb{R}^d \times A \to \mathbb{R} \quad t = 0, \dots, T - 1 \qquad (16)$$

*for each $p \in P$, $z \in \mathbb{R}^d$ and $t = 0, \ldots, T - 1$ one averages them*

$$r_t(p, z, a) = \sum_{p' \in P} \alpha_{p,p'}(a) \mathbb{E}(\tilde{r}_t(p, z, p', W_{t+1}z, a)), \tag{17}$$

*to obtain the usual reward functions, as introduced in the standard setting (3).*

**Approximate solution:** in order to obtain a numerical solution to the above Markov Decision problem, one needs to approximate the true value functions $(v_t^*)_{t=0}^{T-1}$ and the corresponding optimal policies $\pi^* = (\pi_t^*)_{t=0}^{T-1}$. Because all reward and scrap functions are convex in the second variable, the value functions are also convex and they can be approximated by piecewise linear and convex functions.

**Primal solution method:** our approach is based on the observation that, for convex switching systems, the value functions in the backward induction are obtained by applying the following three operations to convex functions:

composition with linear maps, summation (integration), maximization. (18)

To obtain an efficient (approximative) numerical treatment of these operations, the concept of the so-called *sub-gradient envelopes* was suggested in Hinz (2014). A sub-gradient $\nabla_g f$ of a convex function $f : \mathbb{R}^d \to \mathbb{R}$ at a point $g \in \mathbb{R}^d$ is an affine-linear functional supporting this point $\nabla_g f(g) = f(g)$ from below $\nabla_g f \leq f$. Given a finite grid $G = \{g^1, g^2, \ldots, g^m\} \subset \mathbb{R}^d$, the sub-gradient envelope $\mathcal{S}_G f$ of $f$ on $G$ is defined as a maximum of its sub-gradients

$$\mathcal{S}_G f = \bigvee_{g \in G} (\nabla_g f), \tag{19}$$

which provides a convex approximation of the function $f$ from below $\mathcal{S}_G f \leq f$, and it enjoys many useful properties. For our purposes, the following observation is crucial:

*If the function $f$ results from operations in (18) applied to a (large) number $n \in \mathbb{N}$ of convex and piecewise linear argument functions $(f_i)_{i=1}^n$, then $\mathcal{S}_G f$ can be obtained efficiently, unlike the function $f$ itself.*

The reason is that the sub-gradients of $f$ are determined by sub-gradients of argument functions $(f_i)_{i=1}^n$ on grid points only. Thus, all of the operations can be carried out sub-gradient-wise, namely, observe that the summation can be done on the level of sub-gradients

$$\mathcal{S}_G \sum_{i=1}^n f_i = \bigvee_{g \in G} (\sum_{i=1}^n \nabla_g f_i). \tag{20}$$

Furthermore, maximization requires merely sub-gradients of the maximizing function at each grid point

$$\mathcal{S}_G \bigvee_{i=1}^n f_i = \bigvee_{g \in G} \nabla_g f_{\operatorname{argmax}_{j=1}^n f_j(g)}. \tag{21}$$

Finally, the sub-gradient envelope $\mathcal{S}_G f_i(W.)$ of the composition of an argument function $f_i$ with a linear mapping $W$ can be obtained from the composition of all sub-gradients $(\nabla_g f_i)_{g \in G}$ participating in $\mathcal{S}_G f_i$ with $W$ as

$$\mathcal{S}_G f_i(W.) = \bigvee_{g \in G} (\nabla_g f_i)(W.), \quad i = 1, \ldots, n. \tag{22}$$

The crucial point of our algorithm is a treatment of piecewise linear convex functions in terms of matrices. To address this aspect, let us agree on the following notation: given a function $f$ and matrix $F$, we write $f \sim F$ whenever $f(z) = \max(Fz)$ holds for all $z \in \mathbb{R}^d$, and call $F$ a *matrix representative* of $f$. It turns out that the sub-gradient envelope operation

$\mathcal{S}_G$ acting on convex piecewise linear functions, corresponds to a certain row-rearrangement operator $Y_G$ acting on the matrix representatives of these functions, in the sense that

$$f \sim F \quad \Rightarrow \quad \mathcal{S}_G f \sim Y_G[F].$$

Such a row-rearrangement operator $Y_G$ that is associated with the grid

$$G = \{g^1, \ldots, g^m\} \subset \mathbb{R}^d$$

acts on each matrix $F$ with $d$ columns, as follows:

$$(Y_G[F])_{i,\cdot} = F_{\operatorname{argmax}(Fg^i)}, \qquad \text{for all } i = 1, \ldots, m. \tag{23}$$

Let us explain in what sense the properties (20)–(22) are mirrored on the side of matrix representatives. Assume that the piecewise linear and convex functions $(f_i)_{i=1}^n$ are given in terms of their matrix representatives $(F_i)_{i=1}^n$, such that

$$f_i \sim F_i, \quad i = 1, \ldots, n.$$

As a direct consequence of (20)–(22) and the definition (23), it holds that

$$\mathcal{S}_G \left( \sum_{i=1}^n f_i \right) \quad \sim \quad \sum_{i=1}^n Y_G[F_i] \tag{24}$$

$$\mathcal{S}_G \left( \bigvee_{i=1}^n f_i \right) \quad \sim \quad Y_G[\sqcup_{i=1}^n F_i] \tag{25}$$

$$\mathcal{S}_G(f_i(W\cdot)) \quad \sim \quad Y_G[F_i W] \qquad i = 1, \ldots, n, \tag{26}$$

where the operator $\sqcup$ denotes binding matrices by rows. Using the sub-gradient envelope operator, define the double-modified Bellman operator as

$$\mathcal{T}_t^{m,n} v(p,z) = \mathcal{S}_{\mathbf{G}^m} \max_{a \in A} \left( r_t(p,z,a) + \sum_{p' \in P} \alpha_{p,p'}(a) \sum_{k=1}^n v_{t+1}^{(k)} v(p', W_{t+1}^{(k)} z) \right),$$

where the probability weights $(v_{t+1}^{(k)})_{k=1}^n$ correspond to the distribution sampling $(W_{t+1}^{(k)})_{k=1}^n$ of each disturbance matrix $W_{t+1}$. The corresponding backward induction

$$v_T^{m,n}(p,z) \quad = \quad \mathcal{S}_{\mathbf{G}^m} r_T(p,z), \tag{27}$$

$$v_t^{m,n}(p,z) \quad = \quad \mathcal{T}_t^{m,n} v_{t+1}^{m,n}(p,z), \qquad t = T-1, \ldots, 0, \tag{28}$$

yields the so-called double-modified value functions $(v_t^{m,n})_{t=0}^T$. Under appropriate assumptions on increasing grid density and disturbance sampling, the double-modified value functions uniformly converge to the true value functions in (14) on compact sets (see Hinz 2014). Let us present the algorithm from Hinz (2014) for calculating the modified value functions in terms of their matrix representatives:

**Pre-calculations:** given a grid $G^m = \{g^1, \ldots, g^m\}$, implement the row rearrangement operator $Y = Y_{G^m}$ and the row maximization operator $\sqcup_{a \in A}$. Determine a distribution sampling $(W_t^{(k)})_{k=1}^n$ of each disturbance $W_t$ with corresponding weights $(v_t^{(k)})_{k=1}^n$ for $t = 1, \ldots, T$. Given reward functions $(r_t)_{t=0}^{T-1}$ and scrap value $r_T$, assume that the matrix representatives of their sub-gradient envelopes are given by

$$\mathcal{S}_{G^m} r_t(p, \cdot, a) \sim R_t(p,a), \qquad \mathcal{S}_{G^m} r_T(p, \cdot) \sim R_T(p)$$

for $t = 0, \ldots, T - 1$, $p \in P$ and $a \in A$. The matrix representatives of each double-modified value function

$$v_t^{(m,n)}(p, \cdot) \sim V_t(p) \quad \text{for } t = 0, \ldots, T, \, p \in P$$

are obtained via the following matrix-form of the approximate backward induction in (27) and (28):

**Initialization:** start with the matrices

$$V_T(p) = R_T(p), \qquad \text{for all } p \in P.$$

**Recursion:** For $t = T - 1, \ldots, 0$ and for $p \in P$, calculate

$$V_{t+1}^E(p, a) = \sum_{p' \in P} \alpha_{p,p'}(a) \sum_{k=1}^{n} v_{t+1}^{(k)} \mathrm{Y}\left[ V_{t+1}(p') W_{t+1}^{(k)} \right] \tag{29}$$

$$V_t(p) = \bigsqcup_{a \in A} \left( R_t(p, a) + V_{t+1}^E(p, a) \right) \tag{30}$$

This algorithm is depicted in the Algorithm 1.

---

**Algorithm 1:** Value Function Approximation

> **for** $p \in P$ **do**
> > $V_T(p) \sim \mathcal{S}r_T(p, .)$,   $V_T(p) \leftarrow \mathrm{Y}[V_T(p)]$,
> > **for** $a \in A, t = 0, \ldots, T$ **do**
> > > $R_t(p, a) \sim \mathcal{S}r_t(p, ., a)$,   $R_t(p, a) \leftarrow \mathrm{Y}[R_t(p, a)]$
> > 
> > **end**
> 
> **end**
> **for** $t \in \{T - 1, \ldots, 0\}$ **do**
> > **for** $p \in P$ **do**
> > > **for** $a \in A$ **do**
> > > > $V_{t+1}^E(p, a) \leftarrow \sum_{p' \in P} \alpha_{p,p'}(a) \sum_{k=1}^{n} v_{t+1}^{(k)}(k) \mathrm{Y}\left[ V_{t+1}(p') W_{t+1}^{(k)} \right]$
> > > 
> > > **end**
> > 
> > **end**
> > **for** $p \in P$ **do**
> > > $V_t(p) \leftarrow \bigsqcup_{a \in A} \left( R_t(p, a) + V_{t+1}^E(p, a) \right)$
> > 
> > **end**
> 
> **end**

---

Having calculated matrix representatives $(V_t^E)_{t=0}^T$, approximations to expected value functions are obtained as

$$v_{t+1}^E(p, z, a) = \max(V_t^E(p, a)z) \tag{31}$$

$$v_t(p, z) = \max(V_t(p)z) \tag{32}$$

for all $z \in \mathbb{R}^d$, $t = 0, \ldots, T - 1$, $a \in A$ and $p \in P$. Furthermore, an approximately optimal strategy $(\pi_t)_{t=0}^{T-1}$ is obtained for $t = 0, \ldots, T - 1$ by

$$\pi_t(p, z) = \operatorname{argmax}_{a \in A}(r_t(p, z, a) + v_{t+1}^E(p, z, a)), \tag{33}$$

**Dual diagnostics method:** let us now turn to the diagnostics method following Hinz and Yee (2016) whose proof is found in Hinz and Yap (2015). Suppose that a candidate $(\pi_t)_{t=0}^{T-1}$ for approximatively optimal policy is given. To estimate its distance-to-optimality, we

address the performance gap $[v_0^\pi(p_0, z_0), v_0^{\pi^*}(p_0, z_0)]$ at a given starting point $z_0 = Z_0$. For this, we construct random variables $\underline{v}_0^{\pi,\varphi}(p_0, z_0), \bar{v}_0^{\pi,\varphi}(p_0, z_0)$ satisfying

$$\mathbb{E}(\underline{v}_0^{\pi,\varphi}(p_0, z_0)) = v_0^\pi(p_0, z_0) \leq v_0^{\pi^*}(p_0, z_0) \leq \mathbb{E}(\bar{v}_0^{\pi,\varphi}(p_0, z_0)).$$

The calculation of the expectations $\mathbb{E}(\underline{v}_0^{\pi,\varphi}(p_0, z_0))$, and $\mathbb{E}(\bar{v}_0^{\pi,\varphi}(p_0, z_0))$ is realized through a recursive Monte Carlo scheme with variance reduction, which yields approximations to $\mathbb{E}(\underline{v}_0^{\pi,\varphi}(p_0, z_0))$, and $\mathbb{E}(\bar{v}_0^{\pi,\varphi}(p_0, z_0))$, along with appropriate confidence intervals.

For a practical application of the bound estimation, we assume that an approximate solution yields a candidate $(\pi_t)_{t=0}^{T-1}$ for an optimal strategy, as in (33), based on approximations $(v_t)_{t=0}^{T}$ of the value functions from (32).

**Bound estimation:**

(1) Chose a path number $K$ and a nesting number $I \in \mathbb{N}$ to obtain for each $k = 1, \ldots, K$ and $i = 0, \ldots, I$ independent realizations $(w_t^{i,k})_{t=0}^{T}$ of the random variables $(W_t)_{t=0}^{T}$.

(2) Define, for $k = 1, \ldots, K$, the state trajectories $(z_t^k)_{t=0}^{T}$ recursively

$$z_0^k := z_0, \quad z_{t+1}^k = w_{t+1}^{0,k} z_t^k, \quad t = 0, \ldots, T-1$$

and determine all of the realizations

$$\varphi_{t+1}^k(p, a) = \sum_{p' \in P} \alpha_{p,p'}(a) \left( \frac{1}{I} \sum_{i=1}^{I} v_{t+1}(p', w_{t+1}^{i,k} z_t^k) - v_{t+1}(p', z_{t+1}^k) \right). \quad (34)$$

for $t = 0, \ldots, T-1, \quad k = 1, \ldots, K, \, p \in P, a \in A$.

(3) For each $k = 1, \ldots, K$, initialize the recursion at $t = T$ as

$$\bar{v}_T^k(p) = r_T(p, z_T^k), \quad \underline{v}_T^k(p) = r_T(p, z_T^k), \quad p \in P$$

and continue for $t = T-1, \ldots, 0, p \in P$ by

$$\bar{v}_t^k(p) = \max_{a \in A} \left[ r_t(p, z_t^k, a) + \varphi_{t+1}^k(p, a) + \sum_{p' \in P} \alpha_{p,p'}(a) \bar{v}_{t+1}^k(p') \right],$$

$$a_t^k(p) = \pi_t(p, z_t^k)$$

$$\underline{v}_t^k(p) = r_t(p, z_t^k, a_t^k(p)) + \varphi_{t+1}^k(p, a_t^k(p)) + \sum_{p' \in P} \alpha_{p,p'}(a_t^k(p)) \underline{v}_{t+1}^k(p'). \quad (35)$$

(4) Calculate the sample means

$$\frac{1}{K} \sum_{k=1}^{K}, \bar{v}_0^k(p), \quad \frac{1}{K} \sum_{k=1}^{K} \underline{v}_0^k(p)$$

to estimate the performance gap

$$[v_0^\pi(p, z_0), v_0^{\pi^*}(p, z_0)] \quad (36)$$

from above and below, possibly using in-sample confidence bounds.

This technique is depicted in the Algorithm 2 and is usually referred to as *pathwise stochastic control* and it has gained increasing popularity over the recent decades. We refer the interested reader to Hinz and Yap (2015) and the literature cited therein for the technical details. Such a stochastic control exhibits a helpful *self-tuning* property. The closer the value function approximations resemble their true unknown counterparts, the tighter the bounds in (36) and the lower the standard errors of the bound estimates. We provide an application of this technique to the above battery control problem in what follows.

---

**Algorithm 2:** Solution Diagnostics

---

**for** $k = 0, \ldots, K$ **do**

    $z_0^k \leftarrow z_0$

    **for** $t = 1, \ldots, T$ **do**

        **for** $p \in P, a \in A$ **do**

            $\varphi_{t+1}(p,a) = \sum_{p' \in P} \alpha_{p,p'}(a) \left( \frac{1}{I} \sum_{i=1}^{I} v_{t+1}(p', w_{t+1}^{i,k} z_t^k) - v_{t+1}(p', z_{t+1}^k) \right)$

        **end**

        $z_{t+1}^k = w_{t+1}^{0,k} z_t^k$

    **end**

**end**

**for** $k = 1, \ldots, K$ **do**

    **for** $p \in P$ **do**

        $\bar{v}_T^k(p) \leftarrow \underline{v}_T^k(p) \leftarrow r_T(p, z_T^k)$

    **end**

    **for** $t \in \{T-1, \ldots, 0\}$ **do**

        **for** $p \in P$ **do**

            $\bar{v}_t^k(p) \leftarrow \max_{a \in A} \left[ r_t(p, z_t^k, a) + \varphi_{t+1}^k(p, a) + \sum_{p' \in P} \alpha_{p,p'}(a) \bar{v}_{t+1}^k(p') \right]$

            $a_t^k \leftarrow \pi_t(p, z_t^k)$

            $\underline{v}_t^k(p) \leftarrow r_t(p, z_t^k, a_t^k) + \varphi_{t+1}^k(p, a_t^k) + \sum_{p' \in P} \alpha_{p,p'}(a_t^k) \underline{v}_{t+1}^k(a_t^k)$

        **end**

    **end**

**end**

determine estimators $\frac{1}{K} \sum_{k=1}^{K} \bar{v}_0^k(p) \quad \frac{1}{K} \sum_{k=1}^{K} \underline{v}_0^k(p)$

---

## 4. BESS as Stochastic Switching with Linear State Dynamics

Let us construct a model that fulfills all of the assumptions of Section 2, such that the methodology presented in Section 3 becomes applicable. For this, we introduce the four-dimensional uncontrolled state evolution $(Z_t)_{t=0}^{T}$

$$Z_t = [\ 1, \quad Z_t^{(2)}, \quad Z_t^{(3)}, \quad Z_t^{(4)}, \quad Z_t^{(5)}\ ]^{\top}, \quad t = 0, \ldots, T$$

carrying a constant entry in its first component. This is a minor increase of state dimension, allowing to encompass a broad class of dynamics while fulfilling linear (12) restriction. Let us agree that the processes $(\psi_t)_{t=0}^{T}$, $(\underline{\Psi}_{t+1})_{t=0}^{T}$, $(\overline{\Psi}_{t+1})_{t=0}^{T}$, and $(\varepsilon_t)_{t=0}^{T}$ are functions of the components of $(Z_t)_{t=0}^{T}$, as it follows:

$$\begin{aligned} \psi_t &= g^{(2)}(t, Z_t^{(2)}), & \underline{\Psi}_{t+1} &= g^{(3)}(t+1, Z_{t+1}^{(3)}), \\ \overline{\Psi}_{t+1} &= g^{(4)}(t+1, Z_{t+1}^{(4)}), & \varepsilon_{t+1} &= g^{(5)}(t+1, Z_{t+1}^{(5)}), \end{aligned} \tag{37}$$

where the deterministic affine-linear transformations $(g^{(i)}(t, \cdot))_{t=0}^{T}$, with $i = 2, \ldots, 5$, appropriately describe trends and seasonal patterns.

For a numerical case study, let us more specifically address the above framework. First, we suggest modeling the long-term price component as a function of an auto-regressive process. Therefore, consider a sequence $(N_{t+1}^{(2)})_{t=0}^{T}$ of independent standard normal random variables and introduce the auto-regressive state process $(Z_t^{(2)})_{t=0}^{T}$, such that

$$Z_{t+1}^{(2)} = \mu + \phi Z_t^{(2)} + \sigma N_{t+1}^{(2)}, \quad Z_0^{(2)} = z_0^{(2)} \in \mathbb{R}, \tag{38}$$

with parameters $\mu \in \mathbb{R}$, $\sigma \in \mathbb{R}_+$, and $\phi \in [0, 1]$. To embed the evolution (38) into the state process $(Z_t)_{t=0}^T$, recall that the first component is equal to one for $t = 0, \ldots, T$, which allows for the desired linear dynamics

$$
\begin{bmatrix} 1 \\ Z_{t+1}^{(2)} \end{bmatrix} = \begin{bmatrix} 1 & 0 \\ \mu + \sigma N_{t+1}^{(2)} & \phi \end{bmatrix} \begin{bmatrix} 1 \\ Z_t^{(2)} \end{bmatrix} \quad t = 0, \ldots, T - 1. \tag{39}
$$

Other components can be modeled similarly as time dependent affine-linear functions of auto-regressions. For simplicity, we suggest independent identically distributed random variables

$$
Z_{t+1}^{(3)} = N_{t+1}^{(3)}, \quad Z_{t+1}^{(4)} = N_{t+1}^{(4)}, \quad Z_{t+1}^{(5)} = N_{t+1}^{(5)}, \quad t = 0, \ldots, T - 1, \tag{40}
$$

that yield a linear state dynamics (12) with the following disturbance matrices

$$
W_{t+1} = \begin{bmatrix} 1 & 0 & 0 & 0 & 0 \\ \mu + \sigma N_{t+1}^{(2)} & \phi & 0 & 0 & 0 \\ N_{t+1}^{(3)} & 0 & 0 & 0 & 0 \\ N_{t+1}^{(4)} & 0 & 0 & 0 & 0 \\ N_{t+1}^{(5)} & 0 & 0 & 0 & 0 \end{bmatrix} \quad t = 0, \ldots, T - 1.
$$

Here, $(N_t = (N_t^{(i)})_{i=2}^5))_{t=1}^T$ is a sequence of independent multivariate standard normally distributed random variables. For the dynamics (37), the state variables must be scaled and shifted appropriately. The seasonality is reflected by functions

$$
g^{(i)}(t, z) = u_t^{(i)} + s_t^{(i)} z^{(i)} \quad z^{(i)} \in \mathbb{R}, \ t = 0, \ldots, T - 1, \tag{41}
$$

with deterministic shift $u_t^{(i)} \in \mathbb{R}$ and scale $s_t \in [0, \infty]$ coefficients, $i = 2, \ldots, 5$.

To describe the evolution of the controlled part $(P_t)_{t=0}^T$ of the state dynamics, we assume that the finite set $P$ includes battery levels, which are equidistantly spaced with step size $\Delta > 0$ between levels $\underline{p} = \min P$ and $\overline{p} = \max P$. Given the level change function $\ell : P \times A \to P$, the transitions are not random:

$$
\alpha_{p,p'}(a) = \begin{cases} 1 & \text{if } p' = \ell(p, a), \\ 0 & \text{else,} \end{cases} \quad p \in P, a \in A. \tag{42}
$$

Having defined the state evolution $(Z_t, P_t)_{t=0}^T$ and the processes (37) via functions (41) on states, observe that the rewards (10) depend on both the current and the next state, as in (16):

$$
- f(a)\psi_t - \overline{E}_{t+1}(a, p)\overline{\Psi}_{t+1} + \underline{E}_{t+1}(a, p)\underline{\Psi}_{t+1} - c(p, a). \tag{43}
$$

Indeed, due to (37), $\psi_t$ is a function of $Z_t$, while $\overline{E}_{t+1}(a, p)$, $\overline{\Psi}_{t+1}$, $\underline{E}_{t+1}(a, p)$, and $\underline{\Psi}_{t+1}$ are functions of $Z_{t+1}$. Using (17), we transform (43) to the standard form of the reward:

$$
- f(a)\psi_t - \overline{e}_t(a, p)\overline{\psi}_t + \underline{e}_t(a, p)\underline{\psi}_t - c(p, a). \tag{44}
$$

In this equation, the expected surplus $\underline{e}_t(a, p)$ and shortage $\overline{e}_t(a, p)$ of the imbalance are obtained as integrals:

$$
\underline{e}_t(p, a) = \int_0^\infty x \mathcal{N}(f(a) - b(p, a) - u_{t+1}^{(5)}, (s_{t+1}^{(5)})^2)(dx) \tag{45}
$$

and

$$
\overline{e}_t(p, a) = \int_{-\infty}^0 (-x) \mathcal{N}(f(a) - b(p, a) - u_{t+1}^{(5)}, (s_{t+1}^{(5)})^2)(dx), \tag{46}
$$

respectively, for all $p \in P$ and $a \in A$, where $\mathcal{N}(\xi, \varsigma^2)$ denotes the normal distribution with mean $\xi \in \mathbb{R}$ and variance $\varsigma^2 \in \mathbb{R}_+$. Furthermore, $\underline{\psi}_t$ and $\overline{\psi}_t$ are the expectations of $\underline{\Psi}_{t+1}$ and $\overline{\Psi}_{t+1}$ at time $t$, as given by

$$\underline{\psi}_t = u_t^{(3)} \quad \text{and} \quad \overline{\psi}_t = u_t^{(4)}, \quad t = 0, \ldots, T-1. \tag{47}$$

With almost all of the ingredients now in place, we define the reward functions, in accordance to (10), by

$$\begin{aligned} r_t(p, (z^{(1)}, \ldots, z^{(5)}), a) = \\ -f(a)(u_t^{(2)} + s_t^{(2)} z^{(2)}) - \overline{e}_t(p, a) u_t^{(4)} + \underline{e}_t(p, a) u_t^{(3)} - c(p_t, a_t), \end{aligned} \tag{48}$$

for all $a \in A$, $p \in P$, $(z_t^{(1)}, \ldots, z_t^{(5)}) \in \mathbb{R}^5$, and $t = 0, \ldots, T-1$. Finally, let us introduce the last component—the scrap function. Here, we assume that the entire electricity from the BESS can be sold in the long-term market at time $T$:

$$r_T(p_T, (z^{(1)}, \ldots, z^{(5)})) = p\psi_T = p(u_T^{(2)} + s_T^{(2)} z^{(2)}), \tag{49}$$

for $p \in P, (z^{(1)}, \ldots, z^{(5)}) \in \mathbb{R}^5$. With these definitions, we have formalized the optimal management of battery energy storage systems as a stochastic control problem and can address its numerical solution in the next section.

**Remark 4.** *Note that the reward functions (48) and the scrap function (49) only depend on the second component $z^{(2)}$ of the state variable $z = (z^{(1)}, \ldots, z^{(5)})$. That is, modeling the state evolution using linear dynamics can be reduced to the first two components, as in (39).*

## 5. A Numerical Illustration

Consider a BESS with a total capacity of $\chi \in \mathbb{R}_+$ MWh. We assume that the positions $P \subset [0, \chi]$ represent a grid of all feasible battery levels that range from the minimum level $\underline{p} = \min P$ to the maximum level $\overline{p} = \max P$. Such a discretization of battery levels (which are continuous by their physical nature) is a tribute that we have to pay to make our optimal switching approach applicable. However, our the numerical procedures are efficient, and the discretization can be realized at a sufficiently fine granularity. Further, assume that the space of actions is the Cartesian product of two finite sub-spaces:

$$A = \{1, 2, \ldots, \overline{a}^{(1)}\} \times \{1, 2, \ldots, \overline{a}^{(2)}\},$$

with the interpretation that, by taking the action $a = (a^{(1)}, a^{(2)}) \in A$, the retailer chooses a certain safety margin $f^{(1)}(a^{(1)}) \in \mathbb{R}$ through the first action component $a^{(1)}$ and determines, at the same time, a potential battery charge/discharge $f^{(2)}(a^{(2)})$ via the second action component $a^{(2)}$. In fact, we assume that both sets, that of the safety margins

$$\{f^{(1)}(a^{(1)}) : (a^{(1)}, a^{(2)}) \in A\}$$

and that of the charge/discharge

$$\{f^{(2)}(a^{(2)}) : (a^{(1)}, a^{(2)}) \in A\},$$

are represented by discrete grids, which range from their minimum values $\underline{f}^{(1)}$ and $\underline{f}^{(2)}$ to their maximum values $\overline{f}^{(1)}$ and $\overline{f}^{(2)}$, respectively. In this setting, the safety margin function is given by $f(a^{(1)}, a^{(2)}) = f^{(1)}(a^{(1)})$ and the BESS management variable is defined by

$$\ell(p, (a^{(1)}, a^{(2)})) = \arg\min_{p' \in P} |p' - (p + f^{(2)}(a^{(2)}))|,$$

for all $p \in P$ and $(a^{(1)}, a^{(2)}) \in A$. With this variable, the loss due to battery inefficiency is described as in (5).

The state process $(Z_t)_{t=0}^T := (Z_t^{(1)}, Z_t^{(2)})_{t=0}^T$ is modeled as in Section 4 with parameters $\mu \in \mathbb{R}$, $\sigma \in \mathbb{R}_+$, and $\phi \in [0, 1]$. To reflect a trend and a seasonality in the price evolution, we assume

$$\psi_t = g^{(2)}(t, Z_t^{(2)}) = u_t^{(2)} + s_t^{(2)} Z_t^{(2)}, \qquad t = 0, \dots, T,$$

with deterministic coefficients $u_t^{(2)} = -1 + \cos(2\pi t/\tau)$ and $s_t^{(2)} = 1 + \sin^2(2\pi t/\tau)$, where the parameter $\tau > 0$ represents the period length. Figure 3 depicts the state process $(Z_t^{(2)})_{t=0}^T$ and the long-term electricity prices in Euros $(g^{(2)}(t, Z_t^{(2)}))_{t=0}^T$ for the parameters that are listed in Table 1. Further, we obtain the expected short-term prices:

$$\underline{\pi}_t = u_t^{(3)}, \qquad \overline{\pi}_t = u_t^{(4)}, \qquad t = 0, \dots, T-1,$$

where we set $u_t^{(3)} = 5$ and $u_t^{(4)} = 50$ for all $t = 0, \dots, T-1$.

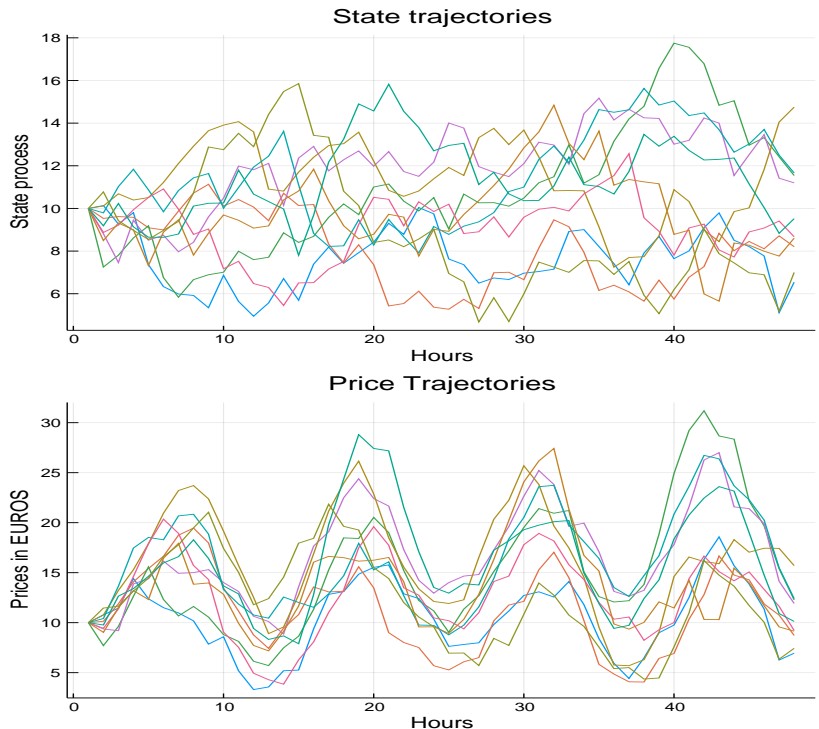

**Figure 3.** The upper and lower plots depict sample paths for the state process $(Z_t^{(2)})_{t=0}^T$ and the corresponding long-term electricity prices $(\psi_t)_{t=0}^T$, respectively.

**Table 1.** The values of the parameters used in the numerical illustration.

| $T$ | $\mu$ | $\phi$ | $\sigma$ | $Z_0^{(2)}$ | $\tau$ | $\kappa$ | $|\mathbf{P}|$ | $|\mathbf{A}|$ | |
|---|---|---|---|---|---|---|---|---|---|
| 48 | 1 | 0.9 | 1 | 10 | 24 | 1 | 21 | $13 \times 9$ | |
| $\underline{p}$ | $\overline{p}$ | $\overline{a}^{(1)}$ | $\overline{a}^{(2)}$ | $\underline{f}^{(1)}$ | $\overline{f}^{(1)}$ | $\underline{f}^{(2)}$ | $\overline{f}^{(2)}$ | $\zeta$ | $\chi$ |
| 0 | 150 | 13 | 9 | $-10$ | 10 | $-10$ | 10 | 1 | 150 |

Finally, we suggest modeling deep discharge costs by the following function:

$$c(p, a) = \eta_1 (1 + \eta_2 p/\chi)^{-1}, \quad p \in P, a \in A,$$

with parameters $\eta_1 \geq 0, \eta_2 > 0$. Note that this function depends on the ratio $p/\chi$, which measures the depth of discharge. Such a function increasingly penalizes the total reward as the battery level approaches zero. To examine the effect of this penalization, we compare the optimal battery levels in Figure 4, which depicts level evolutions under the assumption that the battery starts at the lowest level at time $t = 0$. The bottom plot shows that, in the absence of deep discharge costs ($\eta_1 = 0$), low battery levels are reached routinely. On the contrary, the upper plot shows that battery levels rarely fall below 30% of the total capacity.

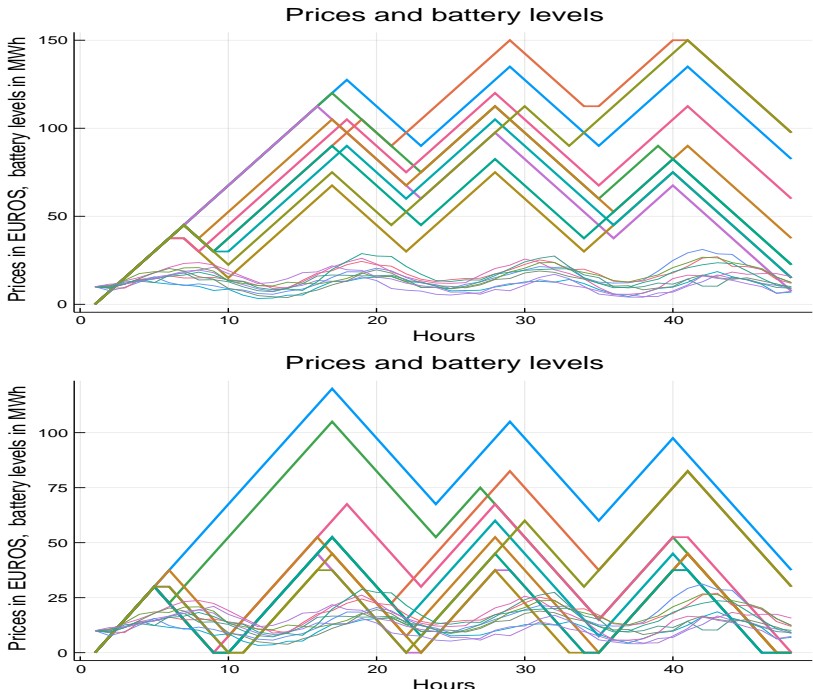

**Figure 4.** Forward electricity prices and battery levels based on the parameter values in Table 1, with $\eta_1 = 100$, $\eta_2 = 15$ (**upper plot**) and with $\eta_1 = 0$ (**lower plot**). The price trajectories, taken from the lower panel of Figure 3, are depicted in the lower part of each panel.

Furthermore, let us illustrate the safety margins in Figure 5. Because there is no significant difference between both graphs, deep discharge costs seem to have a moderate impact on safety margins. In both graphs, we merely see a tendency to buy energy through higher safety margins when electricity prices are low.

Finally, we provide a brief discussion on the value function, which is illustrated in Table 2 and in Figure 6. Each row of Table 2 corresponds to a discretized battery level. The columns "Lower interval" and "Upper interval" include the empirical confidence intervals for the lower and upper estimate of the value function, respectively. This calculation was based on the assumption that the initial electricity price, $\psi_0$, was equal to 10. The confidence bounds, obtained by the diagnostic methods in Hinz and Yap (2015) based on a pathwise dynamic approach, are tight, which certifies the high precision of our solution obtained in terms of the sub-gradient method described in Section 3. Figure 6 depicts the approximate value functions delivered by the sub-gradient method for a range of values of the initial state variable $Z_0^{(2)}$ (represented on the horizontal axis), with different curves standing for different initial battery levels $p_0$. Here, we notice that the value function is increasing in the initial battery level $p_0$ (the energy that is owned at the beginning yields a certain return) and it interacts with the initial state variable, $Z_0^{(2)}$, which is, also, by construction, the initial electricity price $\psi_0$. Recall that a higher price at the beginning causes subsequent prices to be high on average (by the increasing trend of the auto-regressive state process). Therefore, if the battery is well charged at time $t = 0$, then the retailer can sell electricity (within the

time horizon) and obtain a substantial profit; if, on the contrary, the initial level of the battery is low, the retailer must pay more for the initial charge.

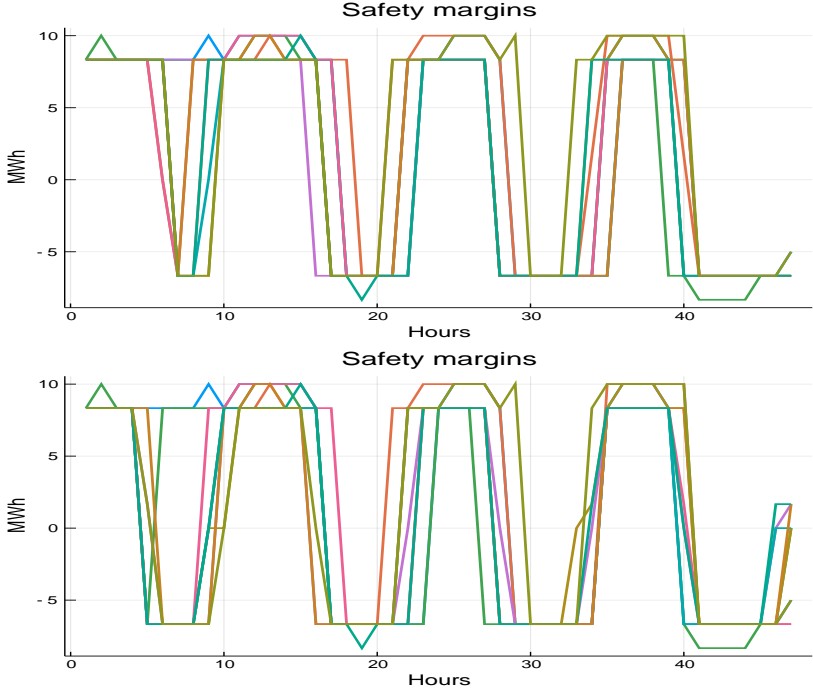

**Figure 5.** Safety margins based on the parameter values in Table 1, with $\eta_1 = 100$, $\eta_2 = 15$ (**upper plot**) and with $\eta_1 = 0$ (**lower plot**). Contrary to Figure 4, the price trajectories, shown in the lower panel of Figure 3, are not reported here to avoid a confusing overlapping with the safety margin plots.

Notice that the numerical results for the value function do not allow a direct interpretation in terms of total revenues. The reason is that the rewards of our model, in (10), do not take the retailer's income from fixed delivery contracts into account (refer to the remark after (10)).

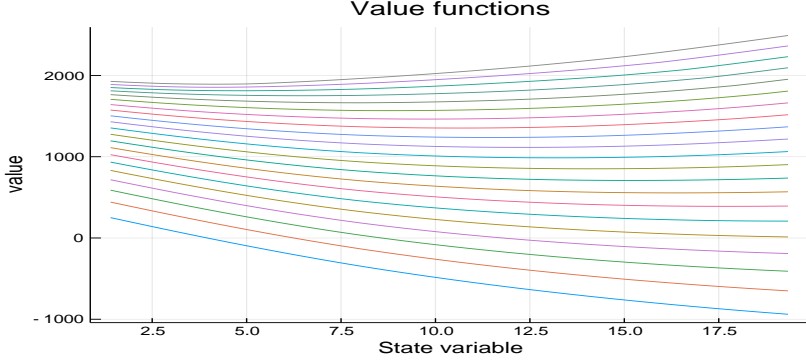

**Figure 6.** Value functions at $t = 0$ under optimal policies. Each curve corresponds to a battery level $p_0 \in P$: the lowest curve is associated with the lowest battery level, 0, the second lowest curve is associated with the second lowest battery level, 8, and so on. Each curve is drawn with respect to a range of values for the state variable $Z_0^{(2)}$. The graphs are drawn with costs of deep discharge ($\eta_1 = 100$ and $\eta_2 = 15$).

Finally, we exemplary elaborate on a typical economic application addressing a stylized investment and capacity allocation problem. In this context, one of the most important questions is to determine the optimal installed capacity and the type of the battery. Having assumed zero costs of deep discharge, Figure 7 depicts the value function, starting with an empty battery, in dependence on storage capacity. Let us refer to this value as the

initial storage value. In line with intuition, a higher storage yields a higher value that is represented by a monotonically increasing concave curve. Because the initial investment in BESS is usually linear in the capacity put in place, this curve could be used to determine the optimal capacity by equating the marginal value of the storage to that of the marginal investment.

**Table 2.** Solution diagnostics based on 50 trajectories (with $\eta_1 = 100$ and $\eta_2 = 15$) and $\psi_0 = 10$.

| Level $p$ (MWh) | Lower Interval | Upper Interval |
| --- | --- | --- |
| 0 | [−484.8457, −484.7002] | [−484.845728, −484.7002] |
| 8 | [−261.5587, −261.4229] | [−261.558277, −261.4219] |
| 15 | [−82.4095, −82.2604] | [−82.409453, −82.2604] |
| 23 | [78.0581, 78.2014] | [78.058582, 78.2023] |
| 30 | [227.8284, 227.9784] | [227.828407, 227.9784] |
| 38 | [370.0146, 370.1593] | [370.015052, 370.1603] |
| 45 | [506.2079, 506.3552] | [506.207923, 506.3552] |
| 53 | [637.4470, 637.6007] | [637.447020, 637.6007] |
| 60 | [764.6185, 764.7616] | [764.620788, 764.7631] |
| 68 | [888.3149, 888.4576] | [888.314914, 888.4576] |
| 75 | [1008.8882, 1009.0220] | [1008.890685, 1009.0233] |
| 83 | [1126.5152, 1126.6511] | [1126.515153, 1126.6511] |
| 90 | [1241.4532, 1241.5858] | [1241.455821, 1241.5871] |
| 98 | [1353.6041, 1353.7311] | [1353.604121, 1353.7311] |
| 105 | [1463.2317, 1463.3635] | [1463.234482, 1463.3646] |
| 113 | [1570.0781, 1570.2042] | [1570.078093, 1570.2042] |
| 120 | [1674.4280, 1674.5548] | [1674.428008, 1674.5548] |
| 128 | [1775.6500, 1775.7734] | [1775.649952, 1775.7734] |
| 135 | [1868.0094, 1868.1358] | [1868.009369, 1868.1358] |
| 143 | [1947.9315, 1948.0580] | [1947.931502, 1948.0580] |
| 150 | [2023.2787, 2023.4052] | [2023.278738, 2023.4052] |

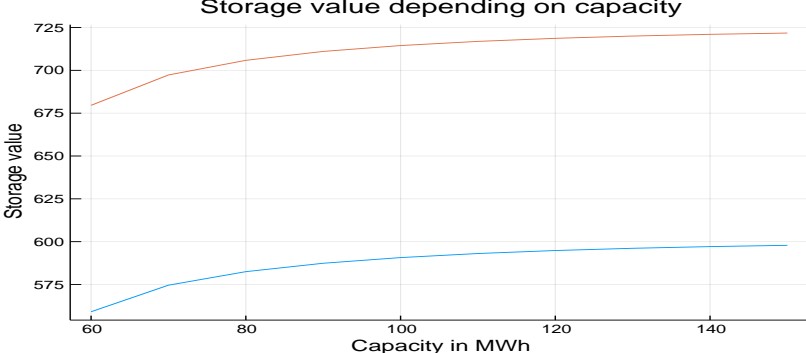

**Figure 7.** Initial value of an empty battery at $\psi_0 = 10$ and for the efficiency parameter $\kappa = 0.97$ (**lower line**) and $\kappa = 1.0$ (**upper line**).

Our numerical experiments suggest that dealing with twenty to fifty equidistant levels shall yield sufficiently precise results (i.e., the numerical outcomes do not change significantly if the granularity becomes finer). However, the discretization of actions (that only yields a finite number of safety margins and battery controls) is a delicate issue. Here, it may be advisable to compare numerical outcomes from several models. Still, our experiments suggest that the optimal strategies are of bang-bang type, meaning that they apply just few extremal (usually largest and smallest) safety margins and battery charging/discharging actions. For this reason, we believe that good results are achievable by small action spaces.

## 6. Conclusions

Battery Energy Storage Systems (BESS) have the potential to change the landscape of future energy generation and trading. However, the existing systems are costly and sensitive to diverse operational issues (such as deep discharge); therefore, a thorough investigation of their optimal management is essential. To engage with this development, we investigate the problem of electricity storage management in the presence of deep discharge costs within a well defined market structure. In particular, we advance a simultaneous optimization of BESS management and energy trading in the presence of the uncertainty resulting from RES generation, demand, and electricity prices. We provide evidence that charging/discharging decisions nicely anticipate changes in electricity prices and avoid, at the same time, deep discharging. Despite the obvious mathematical complications of joining energy trading and BESS management in a stochastic framework, the optimization problem is successfully solved. To this purpose, we adopt a combination of primal and dual schemes that provide an approximately optimal solution with guaranteed accuracy. Our methodology is based on highly time performing computational schemes, whose routines are publicly available. Our approach is also characterized by flexibility, allowing for specifications and adaptations to address real-world problems.

**Author Contributions:** All authors have equally contributed to the paper. All authors have read and agreed to the published version of the manuscript.

**Funding:** This research received no external funding.

**Institutional Review Board Statement:** Not applicable.

**Informed Consent Statement:** Not applicable.

**Data Availability Statement:** No real data has been used.

**Conflicts of Interest:** The authors declare no conflict of interest.

## Note

1    https://github.com/LeePiyachat/rcss (accessed on 17 May 2021).

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
