# Peer review of "A Computational Approach to Sequential Decision Optimization in Energy Storage and Trading"

_jrfm, doi:10.3390/jrfm14060235_

Round 1

Reviewer 1 Report

I wonder if the authors can comment on the pros and cons of the existing commercial  battery storage systems providers, see, for example, 

Compare Battery Storage Systems Providers

https://www.g2.com/categories/battery-storage-systems

Such a comparison will make the paper of interest to practitioners as well.

Author Response

Thank you very much for review, we have updated our work and re-submit in an improved version.

Reviewer 2 Report

The paper is focusing on a quite actual topic. The title is in relevance with the content. The authors applied the appropriate methodological toolset and provided scientific remarkable results. However - and this is my critical point - they didn't arrange a separate methodological chapter where they would describe the methodology of the research. The model is fine actually, so the integrity of the paper would be fine after the recommended upgrade.

Author Response

Thank you very much for review, we have updated our work and re-submit in an improved version.

  • English has been improved and spell checks performed
  • We include now a detailed section on methodology used: Section 3 "Research methodology and solution techniques" guides the reader through a range of quantitative tools which have been applied in this work.  The paper is self-contained now and is much easier to follow. Thank you for the advise.

Reviewer 3 Report

In this paper, the authors discuss the optimization problem in energy trading. The solution is calculated by a numerical approximation method of dynamic programming. Different from other papers about problems of dynamic programming, this paper provides not only a mathematical solution to the problem but also a practical numerical solution. I recommend accept the paper and let many readers are able to apply the method in practice. 

Tiny typo errors:

line 141: {0,1,...,T} ⊂ N instead of {0,1,...,T} ∈ N 

The caption of Figure 3: "The left and right" would better be "The upper and lower"

Author Response

Thank you very much for your work which helped us improving the paper, the English, the style, and the consistency of presentation.